# OPTIMAL UNSUPERVISED DOMAIN TRANSLATION

## ABSTRACT

Unsupervised Domain Translation (UDT) consists in finding meaningful correspondences between two domains, without access to explicit pairings between them. Following the seminal work of *CycleGAN*, many variants and extensions of this model have been applied successfully to a wide range of applications. However, these methods remain poorly understood, and lack convincing theoretical guarantees. In this work, we define UDT in a rigorous, non-ambiguous manner, explore the implicit biases present in the approach and demonstrate the limits of theses approaches. Specifically, we show that mappings produced by these methods are biased towards *low energy* transformations, leading us to cast UDT into an Optimal Transport (OT) framework by making this implicit bias explicit. This not only allows us to provide theoretical guarantees for existing methods, but also to solve UDT problems where previous methods fail. Finally, making the link between the dynamic formulation of OT and CycleGAN, we propose a simple approach to solve UDT, and illustrate its properties in two distinct settings.

Given pairs of elements from two different domains, *domain translation* consists in learning a mapping from one domain to another, linking these paired elements together. If we consider a photograph of a given scene, and an artistic painting of the same scene, we may want to learn the map associating a photograph to paintings describing the same scene, and conversely, for paintings to photographs. A wide range of problems can be formulated as translation, including image-to-image (Isola et al. (2016)), or video-to-video (Wang et al. (2018)), image captioning (Zhang et al. (2016)), natural language translation (Bahdanau et al. (2015)), etc... However, obtaining paired examples can be difficult thus motivating the *unpaired* setting where only samples from both domains are available which allows us to tackle a wider range of problems. A seminal work in this direction has been the CycleGAN model proposed in Zhu et al. (2017a), which has led to extensions for many applications and has given impressive results.

The starting point of this work is to understand and study this successful approach as there remains little theoretical understanding of why these models work. Galanti et al. (2018); Yang et al. (2018) have observed that the approach first introduced in Zhu et al. (2017a) is ill-posed: in most cases, any pairing between samples of both domains – many of which are unwanted pairings – satisfies their objective function. This is in contradiction with the empirical results of the model and shows that there must be an implicit bias towards well-behaved mappings. Galanti et al. (2018) made a first step in this direction by demonstrating that desirable mappings are obtained by networks of minimal *complexity*, relating this notion of complexity to the number of hidden layers of the neural network implementing the mapping. They then show that the number of mappings satisfying the objective is expected to be small[1]. This is indeed a first step but some ambiguity remains: For example, when translating photographs to paintings, instead of preserving the content of the input, we may want to preserve the same color palette. Here, both tasks are in contradiction and *no mapping can solve both problems*: What task will be solved by the CycleGAN model? What is the right notion of complexity to explain this convergence? Can we steer the model toward a certain task?

In this work, we aim to answer these questions and further bridge the gap between practice and theory, by first rigorously and unambiguously defining the UDT problem. Optimal Transport theory then appears as a natural tool to regularize UDT models in this context and its dynamical formulation allows us to build a model which overcomes the shortcomings of CycleGAN-like models.

Our main contributions are the following:

---

[1] Assuming the outputs uniquely define the weights up to invariants, which is not currently shown in the general case.

- Highlighting the need for a more rigorous treatment of the problem of UDT, we reformulate it in a rigorous, non-ambiguous and generic way, showing the theoretical limits of the CycleGAN approach.
- We provide an explanation for the success of CycleGAN, bridging the gap between theory and empirical findings and shedding light on the implicit biases in those models.
- More specifically, this allows us to link UDT and Optimal Transport. We show that any UDT problem can be cast into the Optimal Transport framework and prove that any UDT Task can be solved by selecting the appropriate transportation cost.
- Exploiting the connection between residual networks and dynamical systems, we present a general and flexible approach to solve UDT, based on the Dynamical formulation of Optimal Transport. This allows us to train a steerable one-sided mapping, and obtain the inverse mapping for free after training, along with smooth interpolations between both domains.
- We empirically highlight the implicit bias and limitations of the CycleGAN approach, and demonstrate the potential of the proposed approach in several different scenarios.

## 1 OVERVIEW OF CYCLEGAN-LIKE MODELS

In this section, we describe the main features of the successful CycleGAN approach for solving Unsupervised Domain Translation (UDT).

In the seminal article (Zhu et al. (2017a)), along with many other subsequent works (Lample et al. (2018); Yuan et al. (2018); Chung et al. (2018); Choi et al. (2018)) the presented approach tackles the problem of Unsupervised Domain Translation (UDT), presented as follows: Given samples from two distributions $\alpha$ and $\beta$ on compact domains $\mathcal{A}$ and $\mathcal{B}$, respectively, find neural networks maps $T$ and $S$, such that each network maps one distribution onto the other, while being each other's mutual inverse. More formally, this problem involves minimizing the following loss:

$$\mathcal{L}(T, S, \mathcal{A}, \mathcal{B}) = \mathcal{L}_{\mathrm{coh}}(T, S, \mathcal{A}, \mathcal{B}) + \mathcal{L}_{\mathrm{inv}}(T, S, \mathcal{A}, \mathcal{B})$$

where:

- $\mathcal{L}_{\mathrm{coh}}$ ensures what we will call ***coherence***[2], namely that:

$$T_\sharp \alpha = \beta \quad \text{and} \quad S_\sharp \beta = \alpha$$

- $\mathcal{L}_{\mathrm{inv}}$ ensures what we will call ***inversibility***, or cycle-consistency in the original paper, namely that:

$$S \circ T \stackrel{\alpha-a.s.}{=} \mathrm{id} \quad \text{and} \quad T \circ S \stackrel{\beta-a.s.}{=} \mathrm{id}$$

$\mathcal{L}_{\mathrm{inv}}$ is usually referred to as the *Cycle-Consistency* term, enforced using the $L^1$ norm. $\mathcal{L}_{\mathrm{coh}}$ is a GAN term (Goodfellow et al. (2014)), hence the name of the model, rendering the image distributions indistinguishable from the target distribution, and $T$ and $S$ are often implemented by residual networks (He et al. (2016)).

If the set of *coherent* and *invertible* mappings was reduced to a singleton, this loss would define a well-posed problem with a unique solution. However, this is not the case in any non-trivial setting (Galanti et al. (2018); Benaim et al. (2018)): If the two distributions are discrete and finite with $N$ points each, there are $N!$ such mappings; this number becomes infinite when the distributions have a density w.r.t. the Lebesgue measure. This means that **all mappings from $\alpha$ to $\beta$ satisfying these minimal requirements are optimal** w.r.t. the CycleGAN objective, contradicting empirical results which tend to show that for a certain number of tasks, the model robustly converges toward mappings which give satisfactory results.

## 2 UNSUPERVISED DOMAIN TRANSLATION

In this section, we define rigorously UDT and use this definition to study CycleGAN models theoretically and empirically.

---

[2]The *push-forward* $f_\sharp \rho$ is defined as $f_\sharp \rho(B) = \rho(f^{-1}(B))$, for any measurable set $B$. Said otherwise, coherence means that $T$ maps $\alpha$ to $\beta$ and $S$ does the reverse.

## 2.1 A FORMAL DEFINITION

As seen previously, CycleGan's optimization problem admits as optima all possible coherent and invertible mappings, therefore not well-specifying the problem. This motivates the need for a clear and rigorous definition of an *UDT Task*.

Let $\mathcal{X}_{\alpha,\beta} = \{X | X : \mathcal{A} \rightarrow \mathcal{B}, \ X \ coherent \ \text{and} \ invertible \ w.r.t. \ (\alpha, \beta)\}$ denote the set of all mappings from $\alpha$ to $\beta$. For any given UDT Task, let $\mathcal{T} \subset \mathcal{X}_{\alpha,\beta}$ be the set of all *desirable* mappings for the domain pair $(\alpha, \beta)$. In this case, $\mathcal{T}$ not only depends on the domain pair $(\alpha, \beta)$, but may also vary from one application to another. For instance, in the example task of translating photographs to paintings, we may want all solution mappings $T \in \mathcal{T}$ from photographs to paintings to describe the same underlying reality, and thus to preserve as much as possible the content of the photograph. However, another valid task may be to link photographs to paintings with the same color palette, thus preserving as much as possible the colors of the input, and not necessarily the content of the photograph. As we will show in 2.2 and in the experiments of 4.1, in order to solve UDT in a generic way, a general approach must be able to treat both tasks, and recover desirable mappings for each task.

This clearly demonstrates the necessity of redefining the problem of UDT, in order to treat it in a generic way:

**Definition 2.1** (UDT Task). Consider $\alpha, \beta$, two distributions respectively supported on $\mathcal{A}$ and $\mathcal{B}$ and representing each domain and $\mathcal{T} \subset \mathcal{X}_{\alpha,\beta}$. A *UDT Task* can then be defined as the triple $(\alpha, \beta, \mathcal{T})$ where $\mathcal{T}$ represents the set of all mappings suitable for the considered task[3].

It is now possible to formally define what it means for a UDT Task to be solved by a given approach:

**Definition 2.2** ($(A)$ *solves* UDT Task $(\alpha, \beta, \mathcal{T})$). Let $(A)$ be an approach for UDT, defined by an optimization program $P$ which yields the non-empty set of optima $\mathcal{S} \subseteq \mathcal{X}_{\alpha,\beta}$. A UDT Task $(\alpha, \beta, \mathcal{T})$ is then said to be *solved* by $(A)$ when $\mathcal{S} \subseteq \mathcal{T}$.

In short, the couple of domains $(\alpha, \beta)$ doesn't define a valid problem in itself and there has to be a more specific requirement defining valid mappings.

## 2.2 CYCLEGAN-LIKE MODELS ARE ILL-POSED

From the previous definitions and observations, it directly follows that:

**Proposition 1.** *The only task CycleGAN-based methods solve is* $(\alpha, \beta, \mathcal{X}_{\alpha,\beta})$.

As $\mathcal{X}_{\alpha,\beta}$ is the set of all minimal-requirement mappings from $\alpha$ to $\beta$ and is of infinite size in general, solving this task is trivial, non restrictive, and thus not meaningful in practice. However, as demonstrated by empirical results, the effective task solved by these methods is one that is clearly more restrictive than $(\alpha, \beta, \mathcal{X}_{\alpha,\beta})$, thereby contradicting the former theoretical result. This exhibits the presence of implicit bias in the chosen architectures, hyper-parameters or training methods, assuring the well-posedness the loss does not account for. Recovering these implicit biases, making them explicit is the goal of the following Section 2.3.

It is possible to take one step further and prove the following proposition for any approach of UDT:

**Proposition 2** (No Free Lunch for UDT). *For any approach* $(A)$ *with a non-empty set of optima for two domains* $(\alpha, \beta)$ *not reduced to singletons, there exists a UDT task which it doesn't solve.*

*Proof.* Consider $\mathcal{S}$ as the set of optima given by $(A)$. If $\mathcal{S} = \mathcal{X}_{\alpha,\beta}$, as $\mathcal{X}_{\alpha,\beta}$ has more than one element because $\alpha, \beta$ aren't singletons by hypothesis, there exists $\mathcal{T}_0 \subsetneq \mathcal{X}_{\alpha,\beta}$ and the task $(\alpha, \beta, \mathcal{T}_0)$ verifies $\mathcal{S} \not\subseteq \mathcal{T}_0$ and thus isn't solved by $(A)$. If $\mathcal{S} \neq \mathcal{X}_{\alpha,\beta}$, we take $T \in \mathcal{X}_{\alpha,\beta} - \mathcal{S}$ and $(\alpha, \beta, \mathcal{T}_0 = \{T\})$ isn't solved by $(A)$. $\square$

These results demonstrate that no given method for UDT, in particular methods based on CycleGAN, can solve all UDT Tasks but that *a method has to be tailored for each UDT task*. In Section 3, we present a generalized, more specifically targeted approach for UDT, giving us an *explicit* control over the solutions we converge to, thus allowing us to tackle UDT problems that cannot be solved using previously presented methods.

---

[3]For example, one can define such a set through some qualitative or quantitative property corresponding to what is expected from mappings solving the UDT task.

### 2.3 LOW-DIMENSIONAL EMPIRICAL STUDY OF CYCLEGAN

CycleGAN's associated optimization problem is highly non-convex and challenging, with provably many optimal solutions, as shown in the previous section. In practice, it is solved using SGD-based methods, meaning that the retrieved solutions are highly dependent on the parametrization of the weights, and the choice of initialization and training hyper-parameters. As it is very difficult to study CycleGAN on real datasets, we conduct low-dimensional toy experiments[4]. Most notably, we have observed that the initialization gain $\sigma$, *i.e.* the standard deviation of the weights of the residual network, has a substantial impact on the retrieved mappings.

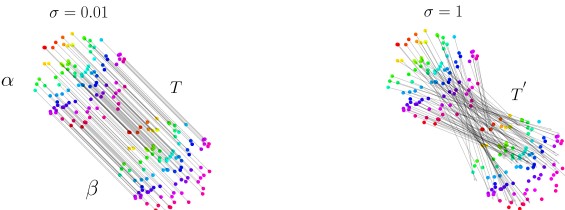

Figure 1: CycleGAN for simple 2D distributions. Small initialization tend to yield simple, ordered mappings (Left), whereas larger initialization yield complex, disordered ones (Right).

In Figure 1, we observe the effect of changing the gain from its original value, $\sigma = 0.01$, to a higher one, $\sigma = 1$. We can see that it does change the obtained mapping from a simple translation aligning the two distributions to a more disorderly one. In other words, higher initialization gains lead to higher energy mappings.

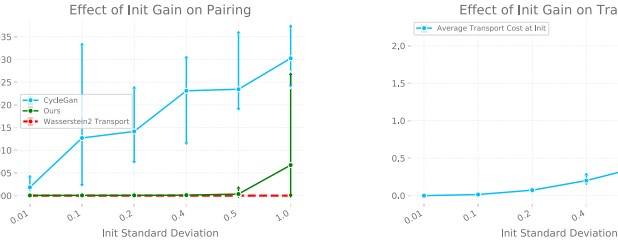

Figure 2: Left: Distance to the minimal transport mapping "*Wasserstein 2 Transport*" as a function of the initialization gain (domains are illustrated in Figure 1). Right: Transport cost of the CycleGAN mapping as a function of init. gain. Metrics are averaged across 5 runs, and the standard deviation is plotted.

A natural characterisation of the *disorder/complexity* of a mapping is the distance between a point $x$ and its image $T(x)$, averaged across points of $\alpha$. *E.g.* using the squared Euclidean distance, this corresponds to the *kinetic* energy of the displacement. As seen in the following section, this quantity, the *transport cost*, is the basis of Optimal Transport theory (Santambrogio (2015)). We use this metric to quantify the effect of initialization gain, Figure 2: On the left, we observe that the larger init. gain becomes, the further CycleGAN's mapping is from the mapping of minimal transport cost *w.r.t.* the Euclidean distance. The right curve confirms and precises this impression: Before training, as the init. gain grows, so does the transport cost of CycleGAN. Note that the variance across runs also increases, *i.e.* the approach yields very different mappings across runs, corroborating the ill-posed nature described in the previous sections.

To summarize, small initialization, those used in practice in CycleGAN, bias the mappings to ones with low Euclidean transport cost, which are less "disorderly", thus giving the simplest alignment of the distributions as output of the model. This works well for many common UDT tasks, where the goal is often to find a "conservative" correspondence between the two domains, which explains the practical success of CycleGAN models despite their ill-posedness. However, for certain tasks, as shown in Section 4, this bias gives the wrong mapping and can't be steered toward the right one.

---

[4]More experiments are available in Section 4.

## 3 Unsupervised Domain Translation as Optimal Transport

Using OT as way to solve UDT arises a very natural thing to do, as for most applications, we wish to preserve input features as much as possible: this is precisely what is given by the OT mapping. Building on the empirical findings of the previous section, we make this implicit bias explicit by formalizing the link between Optimal Transport (OT) and UDT. We outline the main features of OT theory and show how they provide a powerful tool to tackle UDT in the general sense, as posed in Section 2.1. We then present a general and flexible approach to solve UDT, based the dynamical formulation of OT.

### 3.1 From Unsupervised Domain Translation to Optimal Transport

Let $\alpha$ and $\beta$ be two absolutely continuous distributions, and $c$ a cost function. We can then consider the classical Monge formulation of OT, referred to as $(A_c)$:

$$\min_{T} \quad \mathcal{C}(T) = \int_{\mathcal{R}^d} c(x, T(x)) \, \mathrm{d}\alpha(x) \tag{1}$$
$$\text{s.t.} \quad T_\sharp \alpha = \beta$$

We start with a technical definition:

**Definition 3.1** (Twist condition). If $c : \mathcal{A} \times \mathcal{B} \to \mathcal{R}$ is a cost function, it verifies the *Twist condition* if it is differentiable w.r.t. its first variable and, for any $x_0 \in \mathcal{A}$, $y \in \mathcal{B} \to \nabla_{x_0} c(x_0, y)$ is injective.

The following result from OT theory, which is proved in Theorem 1.22 of Santambrogio (2015)[5], not only links OT to UDT, but also ensures that by explicitly minimizing the transportation cost leads to a well-posed optimization problem:

**Theorem 1.** *Let $\alpha$, $\beta$ be absolutely continuous measures. If $c$ verifies the Twist condition, there **exists a unique** couple $(T, S)$ of transformations such that:*

- *$(T, S)$ are coherent and invertible w.r.t. the domain couple $(\alpha, \beta)$;*

- *$\mathcal{C}(T)$ is minimal, and $S$ is the minimal transport from $\beta$ to $\alpha$.*

This result has several important implications: Existence and uniqueness of the minimal transformation are guaranteed given a certain cost $c$, rendering the optimization problem well-posed. Moreover, coherence is not only satisfied for $T$, but also for $S$, along with invertibility for both.

**Remark.** The Twist condition is not very restrictive: Typically, it is verified for any $c(x, y) = h(x - y)$ with $h$ strictly convex.

Building on the previous theorem, there remains the question of knowing whether there exists a cost function adapted for any given UDT task, as defined in Section 2.1. The following answers affirmatively this question for any task which satisfies the preconditions[6]:

**Proposition 3.** *For any UDT task $(\alpha, \beta, \mathcal{T})$, assuming there is at least one map from $\mathcal{T}$ which is differentiable and whose Jacobian is invertible at every point, there exists a cost function $c$ verifying the Twist condition such that $(A_c)$ solves Task $(\alpha, \beta, \mathcal{T})$.*

*Proof.* Consider UDT Task $(\alpha, \beta, \mathcal{T})$, and $T \in \mathcal{T}$, differentiable with invertible Jacobian. Let us define a cost, for instance $c(x, y) = \|T(x) - y\|_2^2$. In this case, $c$ is differentiable w.r.t. its first variable by differentiability of $T$. For $x_0 \in \mathcal{A}$, we then have that:

$$\forall y, \ \nabla_{x_0} c(x_0, y) = 2\,{}^t(\mathrm{Jac}_{x_0} T)(T(x_0) - y)$$

which is clearly injective in $y$ by invertibility of $\mathrm{Jac}_{x_0} T$. Thus $c$ verifies the Twist condition. Moreover, for any transport map $T'$, we have that $\mathcal{C}(T') \geq 0$ and we also have that $\mathcal{C}(T) = 0$ which shows that $T$ is indeed the OT map for $c$ by unicity of the optimum. $\square$

---

[5]It is extended to the Twist Condition setting in Remark 1.24.

[6]This hypothesis cannot be weakened: if the domain $\mathcal{B}$ is convex, for example, a weakening would contradict regularity results of OT maps, see Figalli (2017) for further details.

Note that this existence result is not constructive: the definition of $c$ uses the optimal $T$ we are looking for. In practice, a cost can be constructed using some sort of prior knowledge of the task at hand and **this is a necessary requirement**: without injecting any information about the task —implicitly or explicitly— it is clear that it is impossible to solve any UDT problem in a consistent way. In particular, we advocate for a more rigorous treatment of the problem of UDT, by carefully designing the cost function based on prior information of the task as a way to define the geometry of the problem, as opposed to conducting large hyper-parameter sweeps until the task is empirically solved.

### 3.2 SOLVING UDT USING THE DYNAMICAL FORMULATION OF OPTIMAL TRANSPORT

In the following, we give an equivalent formulation of the Monge problem (1), providing us with a mapping between domains that generalizes the residual architecture of the forward map in CycleGAN. This will also provide a robust computational method to calculate the retrieve the OT mappings in high-dimensional settings while obtaining the inverse without directly parametrizing it. The theoretical framework presented here has been pioneered in Benamou & Brenier (2000) and a detailed modern presentation is given in chapters 4 and 5 of Santambrogio (2015).

**A Dynamical Formulation of OT** In the previous section, $T$ was seen as a static transformation between $\alpha$ and $\beta$. For a wide range of costs, *e.g.* cost of the form $c(x, y) = \|x - y\|_p^p$, there exists a dynamical point of view, similar in intuition and formulation to the equations of fluid dynamics. The general idea is to produce $T$ by using a velocity field $v$ which gradually transports particles from $\alpha$ to $\beta$. The optimal transport map can then be recovered from a path of minimal length, with $v$ solving the optimization problem:

$$\min_v \quad \mathcal{C}^{\text{dyn}}(v) = \int_0^1 \|v_t\|_{L^p(\mu_t)}^p \, \mathrm{d}t \tag{2}$$
$$\text{s.t.} \quad \partial_t \mu_t + \nabla \cdot (\mu_t v_t) = 0, \mu_0 = \alpha, \mu_1 = \beta$$

where $(\mu_t)_{t \in [0,1]}$ is the geodesic path from $\alpha$ to $\beta$ in a measure space (see Figure 6 in the appendix).

For costs of the form[7] $c(x, y) = \|x - y\|_p^p$, **this formulation is equivalent to** (1), meaning that the overall transformation from $\alpha$ to $\beta$ yields the optimal transport cost and is thus the same. The mathematical details are summarized in appendix B.

Directly solving (2) requires solving the continuity equation $\partial_t \mu_t + \nabla \cdot (\mu_t v_t) = 0$, starting from the initial density of $\alpha$. However $\alpha$ is unknown: we only have access to samples and estimating $\alpha$ in high-dimensional spaces is prohibitive. An alternative is to model the trajectories induced by the elements $x \in \mathcal{A}$ the support of $\alpha$, displaced along the vector field $v$, instead of densities. Letting $\phi \colon \mathcal{A} \times [0, 1] \to \mathcal{R}^d$, $(x, t) \mapsto \phi_t^x$ describe the position of elements $x$ of $\alpha$ at time $t$, when they are displaced along $v$ (see figure 6), then the optimization problem can be equivalently written as:

$$\min_v \quad \mathcal{C}^{\text{dyn}}(v) = \int_0^1 \|v_t\|_{L^p((\phi_t)_\sharp \alpha)}^p \, \mathrm{d}t$$
$$\text{s.t.} \quad \partial_t \phi_t^x = v_t(\phi_t^x), \tag{3}$$
$$\phi_0 = \mathrm{id},$$
$$(\phi_1)_\sharp \alpha = \beta$$

where function $\phi_t \colon \mathcal{A} \to \mathcal{R}^d$ is the transport map at time $t$. This problem can be treated as a continuous-time optimal control problem, and can thus be solved using standard techniques Santambrogio (2015). A detailed instance of this approach, used in the experiments of the following section, is given in Section A of the appendix.

**Link with the ResNet implementation of CycleGAN** If $v_k$ corresponds to the residual block at layer $k$ of the residual network defined by $\phi_k^x = \phi_{k-1}^x + v_k(\phi_{k-1}^x)$, taking the continuous time limit recovers the differential equation $\partial_t \phi_t^x = v_t(\phi_t^x)$ (Weinan (2017)). Thus, if we discretize the forward equation in (3) using an Euler numerical scheme, we recover the forward map in the CycleGAN

---

[7] A more general family of costs can be considered in the dynamical setting at the expense of some technicalities, see Figalli (2008).

Table 1: Transport cost of Male to Female translation for different initialization gains.

| Init. Gain | 0.01 | 0.5 | 1.0 | 1.5 |
|---|---|---|---|---|
| CycleGAN | 0.15 | 0.34 | 6.15 | 9.7 |
| Ours | 0.03 | 0.03 | 0.03 | 0.03 |

architecture[8] (cf. A.1 in the appendix). Moreover, using small initialization gains for the network (cf. 2.3) tends to bias $\|v_t\|^p_{L^p((\phi_t^i)_\sharp \alpha)}$ to be small as well, linking latent trajectories of residual networks with these minimal length trajectories: refer to Section C of the appendix, where this is clearly illustrated on a 1d toy UDT Task. Notably, using this continuous formulation, **there is no need to parametrize the inverse**, as it can be simply obtained by taking the velocity field $-v_t$ for the reverse map, automatically dividing the number of necessary parameters by two. Note that continuous interpolations can be easily obtained, saving the intermediate outputs of the forward model.

**Transforming the input and output distributions**    Costs of the form $c(x, y) = \|x - y\|_p^p$ might seem to be too restrictive. However, this apparent difficulty can be circumvented by choosing the right transformation of input and output distribution: Taking two diffeomorphisms $\psi_{(1)}$ and $\psi_{(2)}$, we can replace $(\alpha, \beta)$ by $((\psi_{(1)})_\sharp \alpha, (\psi_{(2)})_\sharp \beta)$ and solve the transformed problem instead, obtain a mapping $T^\psi$ and the solution for the initial problem would be $(\psi_{(2)})^{-1} \circ T^\psi \circ \psi_{(1)}$. For example, one might use an encoder to a well chosen latent space. In this way, it is easy to prove that costs of this form **do not limit the solvability of UDT tasks**.

## 4    EXPERIMENTS

In this section, we describe two sets of experiments, comparing our approach based on OT with the CycleGAN model. Throughout all the experiments, we have used the numerical method based on Equation 3 which practical implementation is described in Section A of the appendix. In particular, in order to stabilize adversarial training, which enforces boundary conditions for both our model and CycleGAN, we have chosen to use an auto-encoder to a lower dimensional latent space. This limits the sharpness of output images but allows to produce consistent and reproducible results which allow meaningful comparisons. Note that in our approach, we only train a one-sided mapping, and retrieve the inverse after training.

### 4.1    MNIST DIGIT SWAP TASK

This toy task uses MNIST data in order to illustrate the fundamental limitation of the CycleGAN family of models and the benefits of using our OT formulation. We construct a dataset consisting of two domains using MNIST digits, for which two different (conflicting) Tasks may be considered. We then show that using our formulation, we are able to solve both Tasks. A detailed overview of the experiment is available in the Appendix, Section D.

### 4.2    CELEBA MALE TO FEMALE

Figure 3 illustrates how our model works for Male to Female translation (forward) and back (reverse) on the CelebA dataset, displaying intermediate images as the input distribution gradually transforms into the target distribution. Note that **no cycle-consistency is being explicitly enforced** here and that the **reverse is not directly parametrized nor trained** but still performs well. The model changes relevant high-level attributes when progressively aligning the distributions but doesn't change non-relevant features (hair or skin color, background,...) which is coherent to what is expected for an optimal map w.r.t. an attractive cost function (here the squared Euclidean one).

Figure 4 and Table 1 compares our model with CycleGAN for various values of the initialization gain hyper-parameter, which effect has been discussed in low dimensions in section 2.3. This confirms our earlier findings: High values of this hyper-parameter make CycleGAN lead to high transport cost,

---

[8]Obviously, other schemes can be used, which would give a different architecture, and some can arguably be more suited for stability reasons but this is beyond the scope of this work.

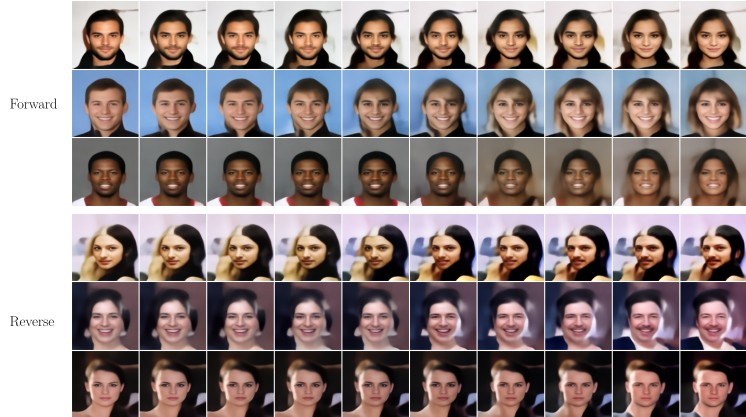

Figure 3: Male to Female translation (top) and the inverse (bottom). Intermediate images correspond to the interpolations provided by the network's intermediate layers. The reverse mapping is obtained by simply inverting the forward network once trained (see text for explanation).

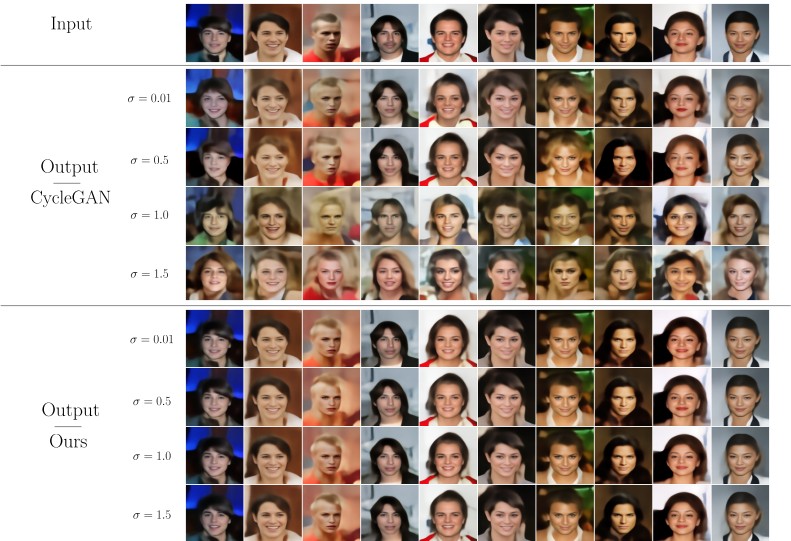

Figure 4: CelebA experiment. Each column associates one input image to its outputs for different models: CycleGAN and our model with different initial gain parameters. Our model gives consistently the same result while CycleGAN converges to different mappings depending on the parameter.

leading to mappings which are not as satisfying for the implicit UDT task being solved here: finding a mapping which transforms men to women and vice-versa while preserving the other features of the mapped image as much as possible. Moreover, training with these high values also induces instability during training, especially in high-dimensional settings, requiring us to map the samples through an encoder before transporting them. Note that our approach is robust to these changes. Finally, note the resemblance between the outputs produced with our approach and CycleGAN with $\sigma = 0.01$.

## 4.3 BIASED CELEBA

In this experiment, we explore the utility of our approach applying our method for solving a UDT Task where the dataset is biased, *e.g.* samples from the dataset do not reflect the true underlying distributions. To this extent, we consider a subset of the CelebA dataset, where domain $\alpha$ and $\beta$ correspond to female faces with **black hair** which are **non-smiling** for $\alpha$, and **smiling** for $\beta$. However, we assume that we do not have access to samples from $\beta$, only samples from an approximate $\beta$, which in this case corresponds to female faces with **blond hair** and **smiling**.

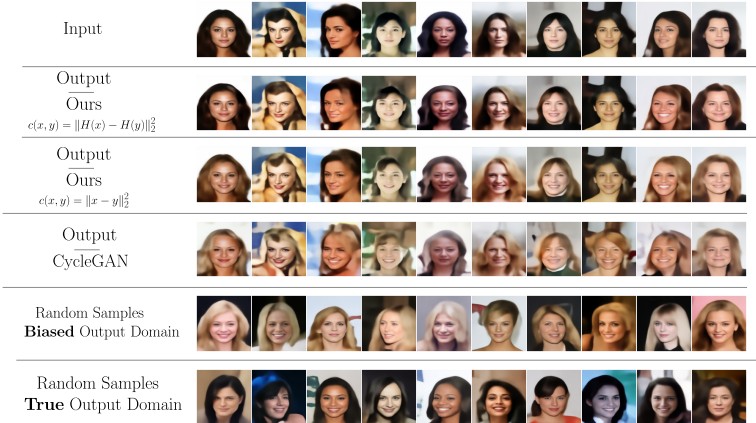

Figure 5: Results for Biased CelebA. We wish to map faces that have the **Non-Smiling** and **Black Hair** attributes to **Smiling**, **Black-Hair** faces, while only accessing **Smiling**, **Blond Hair** faces for the target domain. Our formulation with $c(x,y) = \|H(x) - H(y)\|_2^2$, where $H(x)$ corresponds to the histogram of image $x$, is well adapted for solving this problem.

In this situation, one expects a mapping biased with a quadratic cost to map black-haired non smiling faces to blond smiling ones. This is indeed what happens for the CycleGAN model as we see in figure 5 where we can observe that hair color is modified along with the smile feature. However, if we want to have a mapping which preserves hair color then a stronger, more specific prior has to be added. We achieve this by using **a cost over the color histograms** of the images and the results show that we are thus able to find a mapping which preserves hair color while still finding an otherwise satisfying mapping between the two distributions.

This experiment shows a good example of how to construct a cost: The practitioner defines what needs to be preserved, finds a cost with an appropriate bias (taking into account the structure of the studied datasets) and can then find the desired UDT mappings. It also shows that this method can be used to correct biases in datasets, which were difficult to deal with in the original CycleGAN model.

## 5 DISCUSSION

In this work, we advocate for a more principled treatment of the problem of UDT, by carefully designing the cost function based on prior information of the task as a way to define the geometry of the problem, as opposed to conducting large hyper-parameter sweeps until the task is solved. It is important to note that this formulation does not miraculously solve all UDT problems, since the difficulty now resides in conceiving the right transport cost reflecting the task at hand. This difficulty is symptomatic of a more general difficulty present in unsupervised problems, where one tries to solve a problem without necessarily having a precise knowledge of it.

We believe the link between OT and UDT may provide us with algorithms to solve new UDT problems. Leveraging theoretical and practical advances in OT on UDT may also yield new ways to attack other problems. As we have shown in this work, using this link has allowed us to obtain the inverse mapping along with continuous interpolations of the domains. This may also allow us to tackle *Multi-Domain Translation* (Zhu et al. (2017b)) linking it to the problem of *Multi-Marginal* OT (see Peyré & Cuturi (2018)), or even Many-to-Many Mappings (Almahairi et al. (2018)) which may be tackle using Entropic OT (Peyré & Cuturi (2018)). We leave exploring these links more in depth as future work.

## 6 RELATED WORK

Explaining why practical UDT methods work well when the problem is ill posed has motivated some recent work. Galanti et al. (2018); Benaim et al. (2018), show that coherent mappings are obtained by NNs of minimal *functional complexity*, a notion related to the number of layers. However, this

notion of complexity does not account for problem dependent semantics, and thus it is still not clear which UDT Task this method solves. In our approach, we define a problem-dependent notion of complexity, and solve the associated OT problem. This allows for theoretical results on the solution mapping, *e.g.* results on (non-)existence and (non-)unicity of the solution. They prove the number of such solution mappings is small, assuming that networks with a small number of layers lead to good pairings. Moreover, similarly to us, Galanti et al. (2018); Benaim & Wolf (2017) show that learning a one-sided mapping is possible, however do not obtain the inverse mapping. Others have tried a hybrid approach between paired and unpaired translation Tripathy et al. (2018), which still doesn't solve the problem of ill-posedness.

The dynamical approach presented in Section 3.2 bears similarities with recent work (Chen et al. (2018); Grathwohl et al. (2018)), making use of the link between residual networks and ODEs made in Weinan (2017). Their objective is the design of new neural models and not solving a task like we do. Also different from us, they displace a *known* density (usually Gaussian) along a learned trajectory. This is not possible in our case since both domain distributions are unknown. Gong et al. (2018) uses a variant of the dynamical formulation for this problem but do not make any link with OT nor do they tackle the ill-posedness issue.

In the domain adaptation field, using Optimal Transport to help a classifier extrapolate has been around for some years, e.g. Courty et al. (2015); Damodaran et al. (2018) use a transport cost to align two distributions. The task, although related, is clearly different and so are the methods they develop. In particular they do not consider the dynamical aspect of the transformation process and the link with the dynamics of NNs.

## 7 CONCLUSION

We have defined UDT in a rigorous and general way, and have shown that CycleGAN is biased towards mappings with low transport cost for the squared Euclidean distance. This has allowed us to highlight the approach's limitations, showing both theoretically and empirically that this method cannot solve any given UDT Task. Making the implicit bias present in this model explicit has led us to formulate the problem as an OT problem, which is a very natural way to solve UDT. This approach comes with theoretical guarantees: we have then proven that it is possible to solve any UDT problem within this framework. We make the link between the dynamical formulation of OT and CycleGAN, and propose a simple and robust approach to solve UDT, illustrating its properties in different experimental settings. More generally, we believe that this framework opens many interesting research directions, along with potentially new practical methods to solve existing and novel UDT problems.

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

# A    IMPLEMENTATION DETAILS

## A.1    DISCRETIZATION.

We start by discretizing the forward equation $\partial_t \phi_t^x = v_t(\phi_t^x)$ in time via a $K$ step Euler discretization, starting from $\phi_0^x = x$:

$$\forall x, \ \phi_{(k+1)\Delta t}^x = \phi_{k\Delta t}^x + \Delta t \, v_{k\Delta t}(\phi_{k\Delta t}^x)$$

Here $K$ is the total number of discret steps defining the transformation. Note that by doing so, a residual network architecture is recovered.

In the following, $\Delta t$ will be omitted in $\phi_{j\Delta t}^x$ and $v_{j\Delta t}$. We now replace the unknown distribution $\alpha$ with its empirical $N$ samples counterpart $\frac{1}{N} \sum_{x \in \text{Data}_\alpha} \delta_x$, with $\text{Data}_\alpha$, samples from $\alpha$, we obtain:

$$(\phi_k)_\sharp \alpha \approx \frac{1}{N} \sum_{x \in \text{Data}_\alpha} \delta_{\phi_k^x}$$

corresponding to the empirical distribution induced by the displacement incurred up to step $k$. The cost can now be estimated using the data, summing up the lengths of the trajectories induced by the input data:

$$\mathcal{C}^{\text{dyn}}(v) \approx \frac{\Delta t}{N} \sum_{k=1}^{K} \sum_{x \in \text{Data}_\alpha} \left\| v_k(\phi_k^x) \right\|_p^p$$

Vector field $v_k$ is a function transforming $\phi_k^x$ into $\phi_{k+1}^x$. We parameterize it for each step $k$ using a neural network of parameters $\theta_k$, denoted $v^{\theta_k}$. The optimization problem then amounts at minimizing the norm of residuals, under constraints:

$$\begin{aligned} \min_{\theta} \quad & \mathcal{C}_d(\theta) = \sum_{k=1}^{K} \sum_{x \in \text{Data}_\alpha} \left\| v^{\theta_k}(\phi_k^x) \right\|_p^p \\ \text{s.t.} \quad & \forall x, \ \phi_{k+1}^x = \phi_k^x + \Delta t \, v^{\theta_k}(\phi_k^x), \\ & \phi_0 = \text{id}, \ (\phi_1)_\sharp \alpha = \beta \end{aligned} \tag{4}$$

## A.2    ENFORCING BOUNDARY CONDITIONS.

The forward equation $\phi_{k+1}^x = \phi_k^x + \Delta t \, v^{\theta_k}(\phi_k^x)$ is trivially verified, as is $\phi_0 = \text{id}$. This is not the case for the coherence constraint $(\phi_1)_\sharp \alpha = \beta$ ensuring that input domain $\alpha$ maps to the target domain $\beta$. In order to implement a numerical algorithm, we optimize the Lagrangian associated to (4) (the trivially enforced constraints are not made explicit here), introducing a measure of discrepancy $D$ between output and target domains:

$$\min_{\theta} \quad \mathcal{C}_d(\theta) + \frac{1}{\lambda_i} D\big((\phi_1)_\sharp \alpha, \ \beta\big) \tag{5}$$

where the sequence of Lagrange multipliers $(\lambda_i)_i$ converges linearly to $0$ during optimization, ensuring the constraint is met. This optimization problem is solved using stochastic gradient based techniques. As in most approaches for UDT, $D$ may be implemented using generative adversarial networks Nowozin et al. (2016), or any other appropriate measure of discrepancy between samples, such as kernel distances Gretton et al. (2012), or OT based distances Cuturi (2013).

## A.3    ALGORITHM

Training is done **only for the forward equation**. To obtain the inverse mapping after training, the forward equation is solved in reverse mode. This is immediate and simply amounts at iterating $y_{k-1} = y_k - \Delta t \, v^{\theta_k}(y_k)$, starting from a sample $y_K$ from $\beta$.

We provide below a general description of the algorithm corresponding to our method. Given two unpaired datasets, one first pre-trains an encoder-decoder and uses the obtained encoding as the new data representation to be used to train our model. This learned representation is more suitable than

initial raw data (e.g. images) for representing the semantics and defining relevant OT cost functions. Using the transformed dataset, one then proceeds to training using a mini-batch gradient procedure. First the forward equation is solved, which corresponds to a classical forward pass through the model. The loss in eq. 5 is then computed and a gradient step is performed on all the model parameters. The Lagrangian coefficients are then updated as indicated in section **??** in order to satisfy the constraints of the optimization problem when training ends.

---

**Algorithm 1** Training Procedure

---

**Input:** Dataset of unpaired images $(I_{\mathcal{A}}, I_{\mathcal{B}})$, sampled from $(\alpha, \beta)$,

Initial coefficient $\lambda_0$, decay parameter $d$, initial parameters $\theta$

Pretrain Encoder $E$ and decoder $D$

Make dataset of encodings $(x = E(I_{\mathcal{A}}), y = E(I_{\mathcal{B}}))$

**for** $i = 1, \ldots, M$ **do**

    Randomly sample a mini-batch of $x$, $y$

    Solve forward equation $\phi_{k+1}^x = \phi_k^x + \Delta t \, v^{\theta_k}(\phi_k^x)$ , starting from $\phi_0^x = x$

    Estimate loss $\mathcal{L} = \mathcal{C}_d(\theta) + \frac{1}{\lambda_i} D\big((\phi_1^{\cdot})_{\sharp}\alpha, \, \beta\big)$ on mini-batch

    Compute gradient $\frac{\mathrm{d}\mathcal{L}}{\mathrm{d}\theta}$ backpropagating through forward equation

    Update $\theta$ in the steepest descent direction

    $\lambda_{i+1} \leftarrow \max(\lambda_i - d, 0)$

**end for**

**Output:** Learned parameters $\theta$.

---

## A.4 ARCHITECTURES.

Implementation is performed via DCGAN and ResNet architectures as described below.

For the Encoder, we use a standard DCGAN architecture[9], augmenting it with 2 self-attention layers, mapping the images to a fixed, 128 dimensional latent vector. For the Decoder, we use residual up-convolutions, also augmented with 2 self-attention layers.

For the transportation, we use residual blocks very similar to those in the Resnet architecture proposed in CycleGan[10]: we have 9 residual blocks, each consisting of a linear layer, batch normalization, a non-linearity, and a final linear layer.

The discrepancy $D$ is implemented using generative adversarial networks, although we have observed interesting results with other metrics, *e.g.* using Sinkhorn Distances Cuturi (2013) or MMD Gretton et al. (2012). For the discriminator, we have used a simple MLP architecture of depth 3, consisting of linear layers with spectral normalization, and LeakyReLU($p = 0.2$).

**Hyper-parameters.** We have considered latent dimensions of size 128, the initial coefficient $\lambda_0 = 1$, and the decay factor is set depending on the number of total iterations $M$, so as to be zero on the final iteration. Throughout all the experiments, we use the Adam optimizer with $\beta_1 = 0.5$ and $\beta_2 = 0.999$.

**MNIST Digit Swap Task** Architectures and hyperparameters are the ones presented above. Our dataset is made of $64 \times 64$ images. We have placed resized $32 \times 32$ MNIST 0 and 1 digits, either on the left or on the right, depending on the domain. For training, each training domain is made of 5000 digits of each class. For the test, we use all the 0 and 1 digits available in MNIST.

**Celeba Male to Female Translation.** Architectures and hyper-parameters are the ones presented above. Our dataset is the CelebA dataset, resizing images to $128 \times 128$ pixels, without any additional transformation.

---

[9]`https://github.com/pytorch/examples/tree/master/dcgan`
[10]`https://github.com/junyanz/pytorch-CycleGAN-and-pix2pix`

## B    FROM THE MONGE PROBLEM TO DYNAMICAL OPTIMAL TRANSPORT

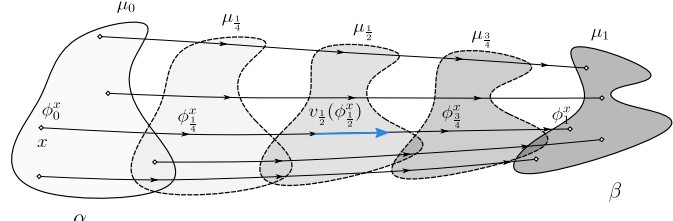

Figure 6: The Figure illustrates successive steps of the dynamic transportation of $\alpha$ to $\beta$ together with the notations used in the text. Each step could for example correspond to a transformation performed by an elementary module of a ResNet.

Instead of directly pushing $\alpha$ to $\beta$ in $\mathcal{R}^d$, it is possible to view $\alpha$ and $\beta$ as points in a space of measures, and consider trajectories from $\alpha$ to $\beta$ in this abstract space. Thus, a way to transport the probability mass from $\alpha$ to $\beta$ is a curve between two points in this space. The curve corresponding to the optimal mapping is then the *shortest* one, in other words it is the *geodesic curve* between the two points.

More formally, let us introduce the *Wasserstein metric space* $\mathbb{W}_p(\mathcal{R}^d)$, *i.e.* the space of absolutely continuous measures of $\mathcal{R}^d$ with finite $p$-th moment endowed with the Wasserstein distance:

$$W_p(\mu, \nu) = \min_{T_\sharp \mu = \nu} \mathcal{C}(T)^{\frac{1}{p}}$$

when costs of the form $c(x, y) = \|x - y\|_p^p$ are considered, for some integer $p \geq 2$. As $\mathbb{W}_p(\mathcal{R}^d)$ is a space of measures, $\alpha$ and $\beta$ are seen as points of this space of measures, and thus, any continuous path linking both distributions defines a gradual transformation from $\alpha$ to $\beta$ and a mapping transporting $\alpha$ to $\beta$.

The following result (from Theorem 5.27 of Santambrogio (2015)) motivates the dynamical formulation of OT:

**Proposition 4.** $\mathbb{W}_p$ *is a geodesic space, meaning that, for any measures* $\mu, \nu \in \mathbb{W}_p$*, there exists a geodesic curve* $(\mu_t)_{t \in [0,1]}$ *between* $\mu$ *and* $\nu$.

Thus, according to this result, finding the optimal mapping between two distributions amounts to finding a curve of minimal length in a certain abstract measure space. However, it still does not provide much in the way of a practically useful algorithm. The following theorem makes a formal link with fluid dynamics and basically states that moving probability masses from one distribution to another is the same as moving fluid densities from one configuration to another under a certain velocity field Santambrogio (2015):

**Theorem 2.** *Given* $\alpha$ *and* $\beta$ *absolutely continuous w.r.t. the Lebesgue measure and* $(\mu_t)_{t \in [0,1]}$ *the geodesic curve with* $\mu_0 = \alpha$ *and* $\mu_1 = \beta$*, we can associate a vector field* $v_t \in L^p(\mu_t)$ *that solves the continuity equation*[11]:

$$\partial_t \mu_t + \nabla \cdot (\mu_t v_t) = 0$$

*with:*

$$W_p^p(\alpha, \beta) = \int_0^1 \|v_t\|_{L^p(\mu_t)}^p \mathrm{dt}$$

In other words, the geodesic curve $(\mu_t)_{t \in [0,1]}$ between both distributions, together with the minimal energy velocity vector field $v$ solve the continuity equation. Moreover, its energy along this path is precisely equal to the Wasserstein distance $W_p^p(\alpha, \beta)$. If this vector field of minimal energy $v$ could be obtained, probability mass could be displaced according to the flow defined by the continuity

---

[11]$\partial_t$ is the partial derivative operator *w.r.t.* variable $t$, and $\nabla\cdot$ the divergence operator *w.r.t.* space.

equation, and the geodesic curve could be retrieved. Thus, we can reformulate the problem as a problem of optimal control, where $v$ is the control variate:

$$\min_v \quad \mathcal{C}^{\mathrm{dyn}}(v) = \int_0^1 \|v_t\|_{L^p(\mu_t)}^p \, \mathrm{d}t$$
$$\text{s.t.} \quad \partial_t \mu_t + \nabla \cdot (\mu_t v_t) = 0, \mu_0 = \alpha, \mu_1 = \beta \tag{6}$$

It is worth noting that this approach not only gives a mapping between the two distributions but it also gives the entire geodesic curve so that smooth interpolations in $\mathbb{W}_p(\mathcal{R}^d)$ can be recovered.

## C  INNER DYNAMICS OF CYCLEGAN

In order to gain a better understanding of CycleGAN's inner dynamics and its relation with low transport maps, we explore the evolution of the input density across the layers of the residual network used for the mapping. We consider the task of UDT, where the input (yellow) and target (green) domains are both 1d Gaussians, with different mean and standard deviation. After training, we plot the evolution of the density across layers in Figure 7 for a small initialization ($\sigma = 0.01$), and a large one ($\sigma = 1.5$). Note that the input samples associated to the input domain are given a color code, and are plotted (under the density), along with densities displaced in time by the optimal transport mapping "Wasserstein 2".

While with the small init. the input samples are displaced to the target in a conservative manner, very similar to the optimal transport map, high initialization can "flip" the original distribution, thus leading to a more chaotic mapping and a higher transport cost. This highlights the link between residual networks used in CycleGAN and optimal transport, made formally justified in Section 3.2.

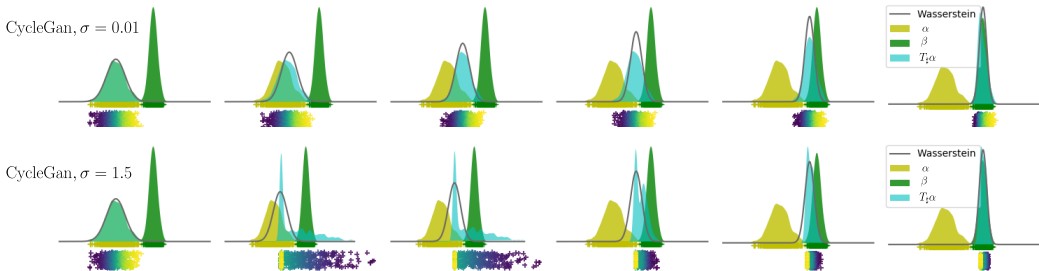

Figure 7: Hidden dynamics of CycleGan, for a simple task of mapping Gaussian $\alpha$ (leftmost yellow density on each plot) to $\beta$ (rightmost green density on each plot), for two init. gains: $\sigma = 0.01$, and $\sigma = 1.5$. A ResNet architecture of 5 blocks is trained on this UDT task. Evolution of the densities across the 5 layers are shown (2nd to 5th plot on each row) (light blue density). The displacement of each point of the empirical input distribution is plotted under the density. We also plot the optimal displacement of the density, *w.r.t.* Wasserstein 2 (blue outlined curve on each plot).

## D    MNIST Digit Swap Task

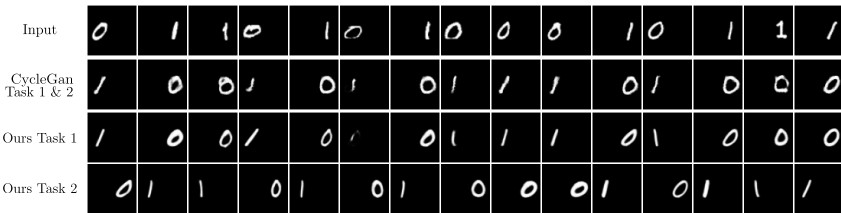

Figure 8: MNIST Digit Swap Task: first row - input, 2nd row - transformation learned by CycleGAN for both tasks, 3rd and 4th row - transformations learned by our model for each task (see text for task definition).

This toy task uses MNIST data in order to illustrate the fundamental limitation of the CycleGAN family of models and the benefits of using our OT formulation.

Table 2: Test-set results on the MNIST Digit Swap Task. Coherence Score: % of success for *"α is mapped on β"*, Score Task 1 & 2: % of success for *"the mapped digit is both correct and at the right position"*.

|  | Coherence | Task 1 | Task 2 |
|---|---|---|---|
| CycleGan | **99.9**% | **99.9**% | 0.00% |
| Ours (Task 1) | **99.9**% | **99.9**% | 0.0% |
| Ours (Task 2) | **99.0**% | 15.42.% | **83.4**% |

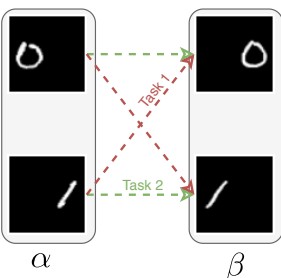

Figure 9: Digit Swap Task.

Let us consider here two domains on MNIST digits: the first domain corresponds to 0s placed on the left of the image and 1s placed on the right (Fig. 8 row 1). For the second, the 0s are placed on the right and 1s on the left.

This defines two tasks (illustrated in Figure 9):
  – **Task 1:** Associate a digit to a digit of the opposite class, keeping style and position unchanged.
  – **Task 2:** Associate a digit to the same digit on the opposite position, preserving the style.

This experiment setup is interesting because the two tasks are conflicting: we cannot solve both at once. Without enforcing any form of prior on the mapping, a coherent mapping (refer to Section 1) is expected to randomly map zeros on the left to zeros on the right, or ones on the left, for instance. However when training CycleGAN, as shown in Figure 8 and Table 2, this is not what happens: the mapping associates digits from one domain to digits in the same position in the other domain in a systematic way, thus *solving Task 1*. This further confirms the low transport bias already shown in section 2.3: our approach, selecting the Euclidean distance as transport cost, yields the exact same results.

As opposed to CycleGAN, our approach can also solve Task 2, by specifying a tailored transport cost (cf. Figure 8 and Table 2). To construct this cost, we first find the components of the latent representations of the images that are the most correlated with the position, training a sparse linear classifier to distinguish the digits position and selecting the features with non-zero weights; We then *turn off* the contribution of these features in the cost function[12]. Implementations details are given in supplementary material, section A. This construction might seem convoluted but this is a normal hurdle: The structure of the solved task has to be embedded in the model, one way or another.

---

[12]More specifically, we use $c(x, y) = \sum_i c_i |x_i - y_i|^2$, where $c_i = 0$ if the classifier's weight associated to component $x^i$ is non-zero, and $c_i = 1$ otherwise.

# E    ADDITIONAL SAMPLES

Forward

Reverse

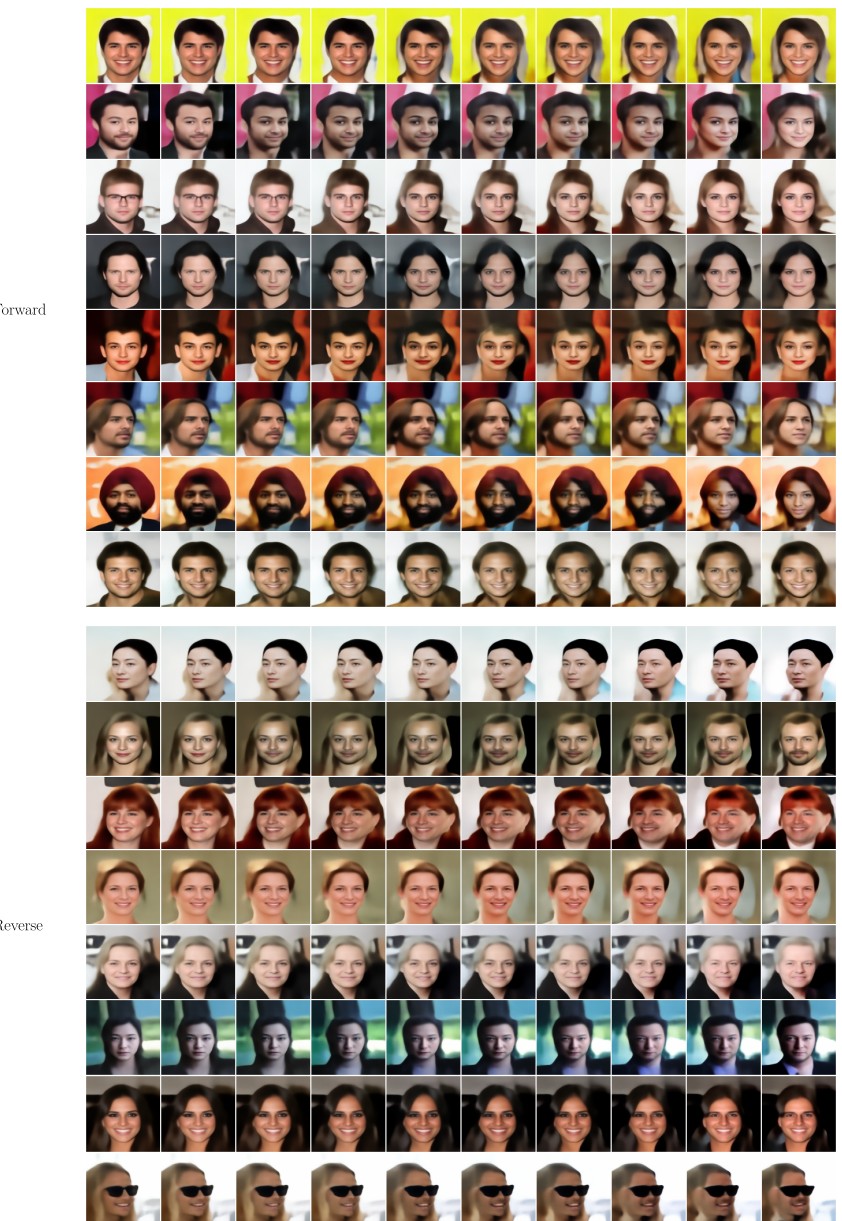

Figure 10: Male to Female, and Back.

