# OpenReview forum: "Optimal Unsupervised Domain Translation"
_ICLR.cc/2020/Conference — Reject_

### Official Review · AnonReviewer1 · 2019-10-18
**Official Blind Review #1**

**Rating:** 3

**Review:**

The paper proposes an analysis of CycleGAN methods. Observing that the class of solutions (mappings) covered by those methods is very large, they propose to restrict the set of solutions by looking at low energy mappings (energy being defined wrt. a dedicated cost). A natural formulation of the associated problem is found in optimal transport (OT) theory. They examine the underlying problem in its dynamical formulation, for which a direct connection can be made with ResNet architecture that are commonly used in cycleGANs. They illustrate these results on simple examples, involving pairing swapped digits from MNIST and celebA male to female examples. As a matter of facts, results presented with the OT formulation are more constant. The main proposition of the paper is that the task at hand can be efficiently coded through the distance (cost) function of OT.

Overall the paper is well written and the proposition is reasonable. Yet some parts seem unnecessary long to me, or bring little information, notably the formalisation of 2.1 and 2.2. The fact that cycleGANs are severely ill-posed problems is well known from the computer vision community. Variants that can include a few paired samples can be found (not exhaustive):
Tripathy, S., Kannala, J., & Rahtu, E. (2018, December). Learning image-to-image translation using paired and unpaired training samples. In Asian Conference on Computer Vision (pp. 51-66). Springer, Cham.
Oe that try to regularize the associated flow:
DLOW: Domain Flow for Adaptation and Generalization Rui Gong, Wen Li, Yuhua Chen, Luc Van Gool; The IEEE Conference on Computer Vision and Pattern Recognition (CVPR), 2019, pp. 2477-2486

In this spirit, I wonder if a comparison with only the vanilla cycleGAN is sufficient to really assess the interest of using the OT formulation of the problem. Notably, in the example of digit swaps, a cost is learnt by finding a representation of the digits that eliminates the importance of position in the representation. Training such a classifier assumes having some labelled data, that could theoretically be paired, and thus making amenable variants of cycleGans that use a few paired samples. In this sense, I think that the paper fails in giving convincing arguments that advocate the use of OT here. As the dynamical formulation is known and already used to learn mappings (  see Trigila, G., & Tabak, E. G. (2016). Data‐driven optimal transport. Communications on Pure and Applied Mathematics, 69(4), 613-648. For instance). Also variants of OT that estimate a Monge mapping could have been included (e.g. V. Seguy, B. B. Damodaran, R. Flamary, N. Courty, A. Rolet, M. Blondel, Large-Scale Optimal Transport and Mapping Estimation, International Conference on Learning Representations (ICLR), 2018.)

As a summary:
Pros:
A nice interpretation of CycleGAN with OT
The paper is fairly well written
Cons:
Overall the quantity of novelties is, in the eyes of the reviewer, somehow limited. At least the contributions should be clarified;
The experimental section is not convincing in explaining why the OT formulation is better than variants of cycleGAN or also other schemes for computing OT than the dynamical formulation

Minor remark:
A reference to Benamou, Brenier 2000 could have been given regarding section 3.2 and the dynamical formulation of OT.


**Experience Assessment:**

I have published one or two papers in this area.

**Review Assessment: Checking Correctness Of Derivations And Theory:**

I assessed the sensibility of the derivations and theory.

**Review Assessment: Checking Correctness Of Experiments:**

I assessed the sensibility of the experiments.

**Review Assessment: Thoroughness In Paper Reading:**

I read the paper thoroughly.

---

> ### Author Response · Authors · 2019-11-15
> **Answer to Review 1**
>
> Dear reviewer, thank you for your review. In the following, we try to alleviate some of your concerns about the practicality and relevance of our approach.
>
>
> Clarification on the contribution:
>
> “Yet some parts seem…”
>
> We believe one of our main contributions in this paper is precisely to analyze the empirical behavior of CycleGAN like models and understand why and how they work despite the ill-posedness of the task as theoretically defined. As far as we know, we are the first to point the link with low energy transformations and to formalize the phenomenon with OT, even though, as stated in the related work section, and as you have rightfully remarked, some similar ideas already exist in the literature [Galanti et al. (2018); Benaim et al. (2018)]. The reviewer's references on similar work are also quoted in to the updated version of the paper, but none of those actually provides well-posedness, theoretical or empirical, in the general case.  Our formalization also has the advantage of providing a natural and flexible regularization which doesn't depend on the architecture or on additional assumptions and which enables us to construct a solution for any given UDT task, not only those involving images.
>
>
>
> Comparison with vanilla CycleGAN and toy digit swap example:
>
> “Notably, in the example…”
>
> There are a few points to be addressed here:
>
> -Giving position and class labels is indeed a form of supervision but isn't equivalent to giving pairings, as there still is the “style” of the number, which is not given as information which we would want to be conserved.  Our use of OT allows to take into account this strictly weaker form of supervision. Our specific tasks were probably not described with enough detail, this will be updated in the revised version.
>
> -This digit swap example task is a toy one: the goal was to show that our model is more expressive than Vanilla CycleGAN and that, given additional information about the solved task, a suitable cost can be constructed.
>
> -However, in general, this is still challenging and some form of supervision is needed in order to do so: This is the meaning of proposition 2 which states that an approach can only be built given a precise specification of the task.
>
> -In particular, the setting with a few pairings is still ill-posed: without additional prior or information, there still would be an infinite (in the continuous case) or hyper-exponential (in the finite discrete case) number of possible mappings for the non paired samples.
>
>
>
> Regarding the choice of a suitable cost:
>
> -As we show in the CelebA experiment, explicitly enforcing a quadratic cost makes the model more robust to the choice of hyper-parameters and there is less variance in the calculated mappings between different training runs. This can be of importance when the training resources are limited. It has to be noted that the quadratic cost already covers the most common situations and in particular all those covered by the CycleGAN model.
>
> - We add to the updated version of the paper another image experiment with a cost over color histograms where the color palette of images is minimally transported instead of their L2 norm. The results show that by using a well chosen cost, we are able to avert the consequences of a bias about hair color in the datasets while the L2 biased mapping fails to do so. This is a good example of how different costs can be leveraged for different tasks (even though quadratic costs are already quite useful in practice) to inject prior knowledge. Looking for other practical applications is then another endeavour and one we will be working on in the future.
>
> -There can be many other problems where different costs could be used in non image areas: for distributions of measures, divergences can be used as ground cost; for distributions of matrices costs over eigenvalues could be more relevant while in medical imaging a specific cost would have to be tailored by field experts, to give just a few examples. The power of the OT approach is that the ground cost can be taken from any cost family as long as it verifies the Twist condition.
>
>
>
> Relevance of the dynamical formulation:
>
> “As the dynamical formulation is known…”
>
> Indeed, the dynamical formulation is used in many other areas and is one of the main causes for the vitality of the OT field. It seemed to be the most straightforward way to generalize CycleGAN given the residual architecture of the mappings in the original paper and gave us reasonably good results but, obviously, other known methods can also be used. There are also many other possibilities to improve this model and extend it to other applications: Entropy and physical regularizations, multi-marginal OT, unbalanced OT,... Looking for concrete applications and for the appropriate specific models to tackle them is part of our current projects and we see this work as a first important exploratory step towards this objective.

---

### Official Review · AnonReviewer3 · 2019-10-21
**Official Blind Review #3**

**Rating:** 6

**Review:**

The paper revisits unsupervised domain translation (UDT) in light of optimal transport. The paper shows that CycleGAN-like models are ill-posed. It redefines UDT tasks using an additional set of suitable mappings. Then the paper redefines UDT problems in the optimal transport framework. Last it proposes an approach to solve UDT problems based on the dynamical formulation of optimal transport. Experiments support the proposed approach.

UDT is a relevant and up-to date problem. The paper helps to clarify some shortcomings of previous approaches and proposes a new solution. The paper is well written. Therefore, in my opinion, the paper should be accepted to ICLR. But, as I am not expert in optimal transport, I would like to have the exact reference of Theorem 1 because I would like to be sure that, in the proof of Proposition 3, the optimum of (A_c) is unique and therefore also satisfies the first item of Theorem 1.

Detailed comments.
* It should be fair to say somewhere that in Zhu et al. (2017a) limits of the approach were already mentioned
* As said before, you should give the exact reference of Theorem 1: which Theorem in Santambrogio (2015). In the proof of proposition 3, you should explain why the minimum of (A_c) is unique and thus corresponds to the minimum in Theorem 1.
* End of Section 3.1. The design of the cost function is left open. This should be made explicit and be discussed somewhere, perhaps in the conclusion.
* I think that there should be a paragraph for the computation of the inverse. This question is considered in different parts of the paper. See for instance the caption of Figure 5. What is the meaning of "inverting the forward network" and to which part of the text does it refer?
* End of Section 4.1. As said before, the design of the cost function is sensitive. Did you have any idea of other cost who would allow to learn the targeted translation without using internal representations?

Typos.
* The notation $T^{\alpha-a.s.} is difficult to read and should be explained
* Beginning of Section 3. "based the dynamical formulation of PT"
* Please check references in texts.  such as "from OT theory Santambrogio (2015)"
* Beginning of Section 3.2., "calculate the retrieve the OT mappings"
* Please give a reference for the dynamical formulation of OT.

**Experience Assessment:**

I have published one or two papers in this area.

**Review Assessment: Checking Correctness Of Derivations And Theory:**

I assessed the sensibility of the derivations and theory.

**Review Assessment: Checking Correctness Of Experiments:**

I assessed the sensibility of the experiments.

**Review Assessment: Thoroughness In Paper Reading:**

I read the paper at least twice and used my best judgement in assessing the paper.

---

> ### Author Response · Authors · 2019-11-15
> **Answer to Review 3**
>
> Dear reviewer, thank you for the extensive review. We will try in the following to address your concerns point by point.
>
> More precise references:
>
> “It should be fair to say somewhere that in Zhu et al. (2017a)...”
>
> Zhu et al. have indeed mentioned some limits  and this will be added to the Related Work section of the updated version of the paper.
>
> “As said before, you should give the exact reference of Theorem 1…”
>
> The exact references for Theorem 1 can be found in [Remark 1.24. + Theorem 1.22, Santambrogio], or [Prop 10.38, Villani]. Note that this result can be generalized in the dynamical formulation, where one posits a cost function for each time t, the twist condition assumption on the “instantaneous cost” also provides optimality and uniqueness [Prop 10.15, Villani].
>
>
>
> Computation of the inverse:
>
> “I think that there should be a paragraph for the computation of the inverse…”
>
> Computation of the inverse is briefly discussed in the appendix, section A.3, however we will provide a more thorough explanation in the revised version of the paper. It consists in solving the differential equation $\dfrac{dx(t)}{dt} = - v(x(t))$ starting from a sample y in $\beta$. This can be intuitively understood as starting from y, and taking the path given by opposite direction $-v(x(t))$ for each point in time $ t \in [0, T]$. In the same way as for the forward mapping, this equation is solved by making use of temporal discretization schemes, e.g. the forward Euler method, which gives us the inverse $x_0$, using $x_{k+1} = x_k - v(x_k)$, with $x_0=y$ sampled from $\beta$, as opposed to $x_{k+1} = x_k + v(x_k)$, starting with $x_0 = x$ sampled from $\alpha$.  Note that other methods for computing the inverse can also be used, which are also based on the dynamical viewpoint of residual networks [Chang, Behrmann], and that those procedures are only done at inference (the backward mapping is not trained nor used during training).
>
>
>
> Design and sensitivity of the cost function:
>
> “End of Section 3.1. The design of the cost function is left open…”
>
> “End of Section 4.1. As said before…”
>
> In the case of images, any differentiable distance or similarity based could be used, e.g. SSIM.
> As we show in the CelebA experiment, explicitly enforcing a quadratic cost makes the model more robust to the choice of hyper-parameters and there is less variance in the calculated mappings between different training runs. This can be of importance when the training resources are limited. It has to be noted that the quadratic cost already covers the most common situations and in particular all those covered by the CycleGAN model.
>
> Moreover,  we add to the updated version of the paper another image experiment with a cost over color histograms where the color palette of images is minimally transported instead of their L2 norm. The results show that, by using a well chosen cost, we are able to avert the consequences of a bias about hair color in the datasets while the L2 biased mapping fails to do so. This is a good example of how different costs can be leveraged for different tasks (even though quadratic costs are already quite useful in practice) to inject prior knowledge.
>
> Although these distances are typical for image domain translation tasks, our formulation provides theoretical guarantees for any cost function in other domains as it  allows us to transport any mathematical object for which we can construct a differentiable cost, and is not restricted to images. For example, we could also transport (distributions of) probability measures, and in this case a natural ground cost function could be the KL divergence, or the Jensen Shannon divergence. We are currently working on projects with more practical applications and, if you have any ideas, we would be happy to discuss them.
>
>
> Minor issues:
>
> "The notation $T^{\alpha-a.s.}$ is difficult to read and should be explained."
>
> For two functions f,g, $f =^{\alpha-a.s.} g$ is equivalent to $f(x) = g(x), \forall x \in \text{support}(\alpha)$. We will provide this definition in the updated version of the paper.
>
> "Please give a reference for the dynamical formulation of OT. "
>
> [Chapter 4 & 5, Santambrogio] is our starting point and provide a rigorous and clear exposition. This will be mentioned in the updated version.
>
> References:
> [Villani]: https://cedricvillani.org/sites/dev/files/old_images/2012/08/preprint-1.pdf
> [Santambrogio]: https://www.math.u-psud.fr/~filippo/OTAM-cvgmt.pdf
> [Chang] : https://aaai.org/ocs/index.php/AAAI/AAAI18/paper/view/16517
> [Behrmann]: http://proceedings.mlr.press/v97/behrmann19a.html

---

### Official Review · AnonReviewer2 · 2019-10-22
**Official Blind Review #2**

**Rating:** 6

**Review:**

Summary:
The paper addresses the ill-posedness of the unsupervised domain translation (UDT) problem. It provides a more structured and rigorous problem definition than previous works (mainly CycleGAN-based), and proposes the theory of optimal transport (OT) as a better framework for solving UDT. The paper provides an interesting link between a dynamical formulation of OT and residual networks, which leads to a practical algorithm for solving OT/UDT. Experiments highlight two main points: 1) CycleGAN are biased towards learning nearly identity mappings, and 2) the OT formulation allows for modelling explicit biases in the learned solution through the design of the cost function.

Strengths & Weaknesses:
  +  The paper addresses an important problem, which as far as I know, is widely known but not properly or explicitly addressed in prior work.
  -  While most definitions are rather intuitive, some are still vague so they cannot be constructive. For example, a UDT task is a subset of all possible mappings which are *desirable* for the given task, but it is not clear how we can exactly define *desirable* mappings.
  -  In addition, it is not clear why the set of all mappings X_{alpha,beta} needs to be constrained to invertible mappings. I see invertibility as only a constraint added by CycleGAN to limit the set of possible learned mappings.

  +  The paper makes an interesting observation that CycleGAN is biased towards simple, and nearly identity mappings (which I believe is the main consequence of small initialization values), which could explain its practical success.
  -  However, the paper needs to emphasize that this is particularly tied to the choice of resnet architectures that is commonly used.

  +  I like the proposed dynamical formulation for solving OT and the link to resnets, which provides an interesting practical algorithm.
  -  The main problem that remains unsolved is how to choose the cost function $c$. The paper acknowledges that, and proposes a specific cost functions for the specific tasks of the experimental section.

  -  While experiments support the main claims of the paper, they are still quite limited and do not really have a clear practical significance. The paper would have been much stronger if the proposed approach solves a more practical problem.

In conclusion, while I think that the practical significance of the proposed approach is rather limited, I think that overall it makes an interesting contribution to the domain of UDT which can be useful for future work.


**Experience Assessment:**

I have published one or two papers in this area.

**Review Assessment: Checking Correctness Of Derivations And Theory:**

I assessed the sensibility of the derivations and theory.

**Review Assessment: Checking Correctness Of Experiments:**

I carefully checked the experiments.

**Review Assessment: Thoroughness In Paper Reading:**

I read the paper thoroughly.

---

> ### Author Response · Authors · 2019-11-15
> **Answer to Review 2**
>
> Dear reviewer, thank you for having taken the time to read our work and for your detailed review. In the following we will try to address your comments and questions point by point, do not hesitate to respond, we would be interested in having your feedback.
>
>
> Notion of desirable mapping:
>
>
> “- While most definitions are rather intuitive…””
>
> Your remark is very relevant: It is indeed difficult in general to define what a *desirable* mapping is. Ideally, the set of desirable mappings we used in the definition would be defined through some quantitative criterion. For example, in the image translation experiments (section 4.2), a desirable mapping should preserve as much information as possible about the face being transformed (hence the relevance of the quadratic constraint). However, there are many situations where it is difficult to construct a precise one and, in practice, practitioners try different mappings until a qualitatively satisfying one is found. In all generality, it is difficult to have a general constructive solution: in Prop 2, Section 2.2, we have shown that there is no algorithm that can solve all UDT Tasks without additional information. This is precisely the strongest motivation for the use of OT: If the user knows how to characterize precisely the mappings solving his UDT task he can find a suitable cost which we know must exist for any given UDT task as we prove in Prop. 3 (with mild additional assumptions). If the characterization is more vague, it becomes more difficult to construct a cost for the task and there has to be some trial and error as for any modelling problem.
>
>
> Importance of invertible mappings:
>
> “-I see invertibility...”
>
> For deterministic mappings (like CycleGAN), if invertibility is not verified on the support of the domains, there is either creation or destruction of mass. In other words, a coherent map as defined in section 1 of the paper is invertible in general which means that invertibility doesn’t really further constrain the solution. We chose to keep this redundancy in order to stay close from the CycleGAN formulation and to stress the fact that we are also looking for a backward mapping which is the inverse of the forward one when we are solving a UDT task.
>
>
> Architecture choice:
>
> “- However, the paper needs to emphasize…”
>
> We studied the ResNet architecture in particular for two reasons: It is the most used one and the similarity of its structure to the discretization of ODEs makes its analysis easier and its generalization to dynamical OT more natural. However, the implicit bias we show is also present in other architectures, e.g. the Unet with its skip connections: The results produced by “good” CycleGAN-like models always seem to have the property of preserving the structure of the input which means that the transformation is a low-energy one.
>
>
>
> Cost definition:
>
> “- The main problem that remains unsolved…”
>
> We do agree that constructing a cost is challenging in general, as we wrote in the paper. However, we think that, while our main focus has been on explaining the empirical success of CycleGAN, the insights we gained through this analysis do have some practical significance:
>
> -As we show in the CelebA experiment, explicitly enforcing a quadratic cost makes the model more robust to the choice of hyper-parameters and there is less variance in the calculated mappings between different training runs. This can be of importance when the training resources are limited. It has to be noted that the quadratic cost already covers the most common situations and in particular all those covered by the CycleGAN model.
>
> -In some situations, the implicit low-energy biais present in CycleGAN might not be the right one when we would like to transport different mathematical objects: e.g. if we were to transport matrices, one would preferably consider minimally displacing the eigenvectors instead of minimally displacing each component as would be done in a trivial extension of CycleGAN. Our approach allows to handle those situations, provided that enough is known to construct a cost; otherwise trial and error is needed but it has to be noted that ground costs can be taken from a large family of functions as the Twist condition is the only necessary property.
>
> “- While experiments support the main claims of the paper…”
>
> - We add to the updated version of the paper another image experiment with a cost over color histograms where the color palette of images is minimally transported instead of their L2 norm. The results show that by using a well chosen cost, we are able to avert the consequences of a bias about hair color in the datasets while the L2 biased mapping fails to do so. This is a good example of how different costs can be leveraged for different tasks (even though quadratic costs are already quite useful in practice) to inject prior knowledge. Looking for other practical applications is then another endeavour and one we will be working on in the future.

---

### Author Response · Authors · 2019-11-15
**Summary describing main changes and additional experiments**

First of all, we would like to kindly thank the reviewers, who have all taken the time to give us useful feedback on the paper by writing detailed comments about different aspects of our work. We have answered each reviewer individually and will outline here the main changes in the revised version of our work:

To summarize:

* An issue pointed out by the reviewers was that the experiments did not demonstrate the generic nature of the approach, by providing examples of useful cost functions. An additional experiment demonstrating the advantage of our approach in the context of biased datasets has been added,

* Additional explanations for specific parts of the paper (e.g. computation of the inverse) will be added,

* References describing related work provided by the reviewers will have been and will be added,

* Detailed citations, describing the exact position in the books have be added,

* Corrections for typos and other small errors will be provided.

---

### Decision · Program_Chairs · 2019-12-19

**Decision:**

Reject

**Comment:**

The paper examines the problem of unsupervised domain translation. It poses the problem in a rigorous way for the first time and examines the shortcomings of existing CycleGAN-based methods. Then the authors propose to consider the problem through the lens of Optimal Transport theory and formulate a practical algorithm.

The reviewers agree that the paper addresses an important problem, brings clarity to existing methods, and proposes an interesting approach / algorithm, and is well-written. However, there was a shared concern about whether the new approach just moves the complexity elsewhere (into the design of the cost function). The authors claim to have addressed in the rebuttal by adding an extra experiment, but the reviewers remained unconvinced.

Based on the reviewer discussion, I recommend rejection at this time, but look forward to seeing the revised paper at a future venue.